# LEARNING TO MAP FOR
# ACTIVE SEMANTIC GOAL NAVIGATION

**Georgios Georgakis\*[1], Bernadette Bucher\*[1], Karl Schmeckpeper[1], Siddharth Singh[2], Kostas Daniilidis[1]**
[1]University of Pennsylvania, [2]Amazon
{ggeorgak, bucherb, karls}@seas.upenn.edu,hartsid@amazon.com, kostas@cis.upenn.edu

## ABSTRACT

We consider the problem of object goal navigation in unseen environments. Solving this problem requires learning of contextual semantic priors, a challenging endeavour given the spatial and semantic variability of indoor environments. Current methods learn to implicitly encode these priors through goal-oriented navigation policy functions operating on spatial representations that are limited to the agent's observable areas. In this work, we propose a novel framework that actively learns to generate semantic maps outside the field of view of the agent and leverages the uncertainty over the semantic classes in the unobserved areas to decide on long term goals. We demonstrate that through this spatial prediction strategy, we are able to learn semantic priors in scenes that can be leveraged in unknown environments. Additionally, we show how different objectives can be defined by balancing exploration with exploitation during searching for semantic targets. Our method is validated in the visually realistic environments of the Matterport3D dataset and show improved results on object goal navigation over competitive baselines.

## 1 INTRODUCTION

What enables biological systems to successfully navigate to semantic targets in novel environments? Consider the example of a dog whose food tray at its own house is situated next to the fridge. Upon entering a new house for the first time, the dog will look for its food tray next to the fridge, even though the new house can largely differ in appearance and layout. This is remarkable, as it suggests that the dog is able to encode spatial associations between semantic entities that can be leveraged when trying to accomplish a navigation task. Humans exhibit the same skills in similar scenarios, albeit more nuanced, since given existing observations we can consciously choose to trust our prior knowledge over the semantic structure of the world, or to continue exploring the environment. In other words, if we have a partial view of a room containing an oven, we can infer that a fridge most likely exists in the unobserved space. In addition, if we are trying to reach the sofa, then we can infer with high certainty that it will be located in a different room. This implies that we have internal mechanisms for quantifying the uncertainty of inferred information from unobserved spaces, which guides our decision making process.

Inspired by these observations, in this work, we study the problem of object goal navigation for robotic agents in unseen environments and propose an active learning method for encoding semantic priors in indoor scenes. Our approach involves learning a mapping model than can predict (hallucinate) semantics in unobserved regions of the map containing both objects (e.g. chairs, beds) and structures (e.g. floor, wall), and during testing uses the uncertainty over these predictions to plan a path towards the target. Contrary to traditional approaches for mapping and navigation (Cadena et al., 2016) (i.e. SLAM) where the focus is on building accurate 3D metric maps, our uncertainty formulation is designed to capture our lack of confidence about whether a certain object exists at a particular location. This results in a much more meaningful representation, suitable for target-driven tasks.

Recently, learned approaches to navigation have been gaining popularity, where initial efforts in addressing target-driven navigation focused on end-to-end reactive approaches that learn to map pixels directly to actions (Zhu et al., 2017; Mousavian et al., 2019). These methods do not have an

---

\* Denotes equal contribution.

explicit representation of the environment and tend to suffer from poor generalization. To remedy this issue, most current methods learn a map representation that enables the encoding of prior information about the geometry and semantics of a scene, acting as an episodic memory (Chaplot et al., 2020b;a; Gupta et al., 2017; Georgakis et al., 2019). However, maps created by these methods are restricted to contain information only from areas that the agent has directly observed, which led to the introduction of spatial prediction models that either anticipate occupancy (Santhosh Kumar Ramakrishnan & Grauman, 2020) or room layouts (Narasimhan et al., 2020) beyond the agent's field of view and demonstrated improved performance on navigation tasks. Our work differs from these methods in three principled ways: 1) We formulate an active training strategy for learning the semantic maps, 2) we exploit the uncertainty over the predictions in the planning process, and 3) in contrast to predicting occupancy, our model tackles a harder problem which requires learning semantic patterns (e.g. tables surrounded by chairs).

In this work we introduce *Learning to Map* (L2M), a novel framework for object-goal navigation consisting of two parts. First, we actively learn an ensemble of two-stage segmentation models by choosing training samples through an information gain objective. The models operate on top-down maps and predict both occupancy and semantic regions. Second, we estimate the model uncertainty through the disagreement between models in an ensemble from Pathak et al. (2019); Seung et al. (1992), and show its effectiveness in defining objectives in planners to actively select long-term goals for semantic navigation. In addition, we investigate different information gain objectives during active training and illustrate how the use of model uncertainty can balance exploration with exploitation in finding semantic targets. Our proposed approach demonstrates improved success rates on the object-goal navigation task over competitive baselines in the Matterport3D (Chang et al., 2017) dataset using the Habitat (Savva et al., 2019) simulator.

## 2 RELATED WORK

**Semantic SLAM.** Classical approaches for navigation focus on building 3D representations of the environment before considering downstream tasks (Cadena et al., 2016). While these methods are typically geometric, several SLAM methods have attempted to associate semantic information to the reconstructed geometric map, mainly at the object-level (Salas-Moreno et al., 2013; Yang & Scherer, 2019; McCormac et al., 2018; Bowman et al., 2017; Kostavelis & Gasteratos, 2015). For example, in McCormac et al. (2018) instance segmentations predicted by Mask R-CNN are incorporated to facilitate per-object reconstructions, while the work of Bowman et al. (2017) proposes a probabilistic formulation to address uncertain object data association. However, SLAM systems rarely consider active exploration as they are not naturally compatible with task-driven learnable representations from deep learning architectures that can encode semantic information. Other recent works (Katsumata et al., 2020; Cartillier et al., 2021) have sought to build 2D semantic maps and focused either on semantic transfer of a global scene or assumed the environments were accessible before-hand. In contrast, our proposed approach tackles the object goal task in unknown environments by actively learning how to predict semantics in both observed and unobserved areas of the map around the agent.

**Learning based navigation methods.** There has been a recent surge of learning based methods (Zhu et al., 2017; Mousavian et al., 2019; Gupta et al., 2017; Chen et al., 2020; Chaplot et al., 2020b; Fang et al., 2019; Yang et al., 2019; Ye et al., 2021; Zhang et al., 2021; Chattopadhyay et al., 2021) for indoor navigation tasks  (Anderson et al., 2018; Batra et al., 2020; Das et al., 2018), propelled by the introduction of high quality simulators such as Gibson (Xia et al., 2018), Habitat (Savva et al., 2019), and AI2-THOR (Kolve et al., 2017). Methods which use explicit task-dependent map representations (Parisotto & Salakhutdinov, 2018; Gupta et al., 2017; Chaplot et al., 2020b;a; Georgakis et al., 2019; Gordon et al., 2018; Mishkin et al., 2019) have shown to generalize better in unknown environments than end-to-end approaches with implicit world representations. For example, in Gupta et al. (2017) a differentiable mapper learns to predict top-down egocentric views of the scene from RGB images, followed by a differentiable planner, while in Chaplot et al. (2020a) Mask R-CNN is used to build a top-down semantic map of the scene used by a learned policy that predicts goal locations in the map. More conceptually similar to our method, are approaches that attempt to encode semantic or layout priors by learning to predict outside the field-of-view of the agent (Santhosh Kumar Ramakrishnan & Grauman, 2020; Liang et al., 2021; Narasimhan et al., 2020; Georgakis et al., 2022). In contrast to all these works we formulate an active, target-independent strategy to predict semantic maps and define goal selection objectives.

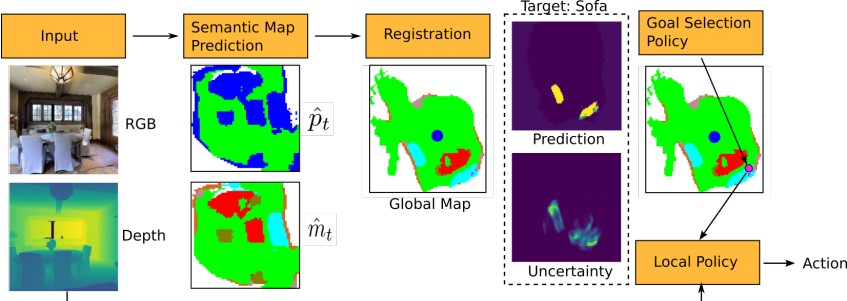

Figure 1: Overview of our approach. We select goals based on the predicted probability of a location containing our target and the uncertainty of the prediction. Note that in this particular example, the sofa to the right of the agent is not visible, but it is correctly predicted by our method.

**Uncertainty Estimation.** Many recent works estimate uncertainty of deep learning models (Gal, 2016; Abdar et al., 2021). In some cases, these uncertainty estimates can be used as objectives for active exploration (Sekar et al., 2020; Pathak et al., 2019) since maximizing epistemic uncertainty is used as a proxy for maximizing information gain (Seung et al., 1992; Pathak et al., 2019). Uncertainty estimates of deep learning models able to be used as objectives for active exploration fall into two categories: Bayesian neural networks and ensembles. Lakshminarayanan et al. (2017); Gawlikowski et al. (2021) found the multiple passes required by Bayesian methods for every prediction can result in a higher computational cost than small ensembles. In addition, ensembles are simpler to implement with little hyperparameter tuning in comparison to Bayesian methods, leading us to focus our work on ensemble based approaches. We tested estimating epistemic uncertainty with entropy (Shannon, 1948), an approximation of model information gain using predictive entropy (BALD) (Houlsby et al., 2011), and variance between model outputs (Seung et al., 1992). We found maximizing the variance yielded the best performance as an objective to actively fine-tune our map prediction models. This training procedure is similar to the active learning approaches leveraged during training presented by Bucher et al. (2021); Chaplot et al. (2020c); Sener & Savarese (2018). We also use our epistemic uncertainty estimate to construct confidence bounds for our estimated probability distribution which we use to select goals for target-driven navigation at test time. Both lower (Galichet et al., 2013) and upper (Auer et al., 2002) confidence bound strategies for balancing exploration, exploitation, and safety have been previously proposed in the multi-armed bandit literature and extended for use in MDPs (Azar et al., 2017) and reinforcement learning (Chen et al., 2017).

## 3 APPROACH

We present a new framework for object-goal navigation that uses a learned semantic map predictor to select informative goals. In contrast to prior work (Santhosh Kumar Ramakrishnan & Grauman, 2020; Chaplot et al., 2020a), we leverage the predictions outside the field-of-view of the agent to formulate uncertainty-based goal selection policies. Furthermore, we actively collect data for training the map predictor and investigate different information gain objectives. Due to our goal selection policy formulation, our method does not need to be trained specifically to predict goals for every target object, enabling a target-independent learning of the semantic priors. Our method takes as input an RGB-D observation, and predicts semantics in the unobserved areas of the map. This is followed by goal selection based on the estimated uncertainty of the predictions. Finally a local policy is responsible for reaching the goal. An overview of our pipeline can be seen in Figure 1.

### 3.1 SEMANTIC MAP PREDICTION

We describe a method for learning how to map by predicting the semantic information outside the field of view of the agent. We emphasize that this goes beyond traditional mapping (i.e. accumulating multiple views in an agent's path) as it relies on prior information encoded as spatial associations between semantic entities in order to hallucinate the missing information. Motivated by the past success of semantic segmentation models in learning contextual information (Zhang et al., 2018; Yuan et al., 2020), we formulate the semantic map prediction as a two-stage segmentation problem.

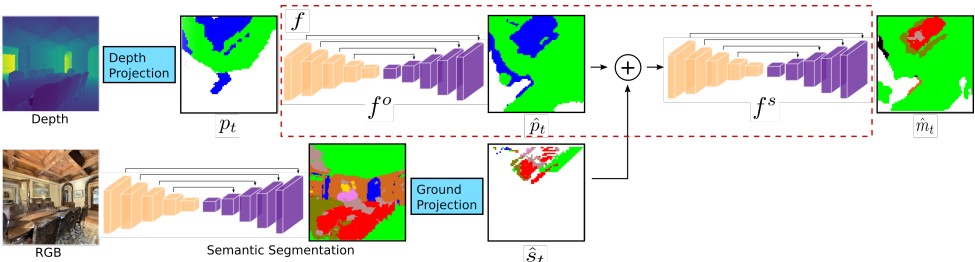

Figure 2: Overview of our semantic map predictor approach for a single time-step. The model predicts the top-down egocentric semantics of unobserved areas in a two-step procedure. First, the occupancy $\hat{p}_t$ is predicted, which is concatenated with a ground-projected semantic segmentation of the RGB observation before producing the final output $\hat{m}_t$. Note that our model learns to predict semantically plausible maps (i.e. chairs surrounding a table) as shown in this example.

Our method takes as input an incomplete occupancy region $p_t \in \mathbb{R}^{|C^o| \times h \times w}$ and a ground-projected semantic segmentation $\hat{s}_t \in \mathbb{R}^{|C^s| \times h \times w}$ at time-step $t$. The output is a top-down semantic local region $\hat{m}_t \in \mathbb{R}^{|C^s| \times h \times w}$, where $C^o$ is the set of occupancy classes containing *unknown*, *occupied*, and *free*, $C^s$ is the set of semantic classes, and $h$, $w$ are the dimensions of the local crop. To obtain $p_t$ we use the provided camera intrinsics and depth observation at time $t$ to first get a point cloud which is then discretized and ground-projected similar to Santhosh Kumar Ramakrishnan & Grauman (2020). To estimate $\hat{s}_t$ we first train a UNet (Ronneberger et al., 2015) model to predict the semantic segmentation of the RGB observation at time $t$. All local regions are egocentric, i.e., the agent is in the middle of the crop looking upwards. Each spatial location in our map (cell) has dimensions $10cm \times 10cm$.

The proposed two-stage segmentation model predicts the hallucinated semantic region in two stages. First, we estimate the missing values for the occupancy crop in the unobserved areas by learning to hallucinate unseen spatial configurations based on what is already observed. Second, given predicted occupancy, we predict the final semantic region $\hat{m}_t$. These steps are realized as two UNet encoder-decoder models, $f^o$ that predicts in occupancy space $\hat{p}_t = f^o(p_t; \theta^o)$, and $f^s$ that predicts in semantic space $\hat{m}_t = f^s(\hat{p}_t \oplus \hat{s}_t; \theta^s)$, where $\hat{p}_t$ is the predicted local occupancy crop which includes unobserved regions, $\oplus$ refers to the concatenation operation, and $\theta^o$, $\theta^s$ are the randomly initialized weights of the occupancy and semantic networks respectively. The image segmentation model is trained independently and its ground projected output $\hat{s}_t$ conditions $f^s$ on the egocentric single-view observation of the agent. The model is trained end-to-end using pixel-wise cross-entropy losses for both occupancy and semantic classes and predicts a probability distribution over the classes for each map location. We assume that ground-truth semantic information is available such that we can generate egocentric top-down occupancy and semantic examples. This combined objective incentivizes learning to predict plausible semantic regions by having the semantic loss backpropagating gradients affecting both $f^o$ and $f^s$. Also, performing this procedure enables the initial hallucination of unknown areas over a small set of classes $C^o$, before expanding to the more difficult task of predicting semantic categories $C^s$ which includes predicting the placement of objects together with scene structures such as walls in the map. An overview of the semantic map predictor can be seen in Figure 2. During a navigation episode, the local semantic region is registered to a global map which is used during planning. Since we predict a probability distribution at each location over the set of classes, the local regions are registered using Bayes Theorem. The global map is initialized with a uniform prior probability distribution across all classes.

## 3.2 UNCERTAINTY AS AN OBJECTIVE

A key component of a robotic system is its capacity to model what it does not know. This ability enables an agent to identify failure cases and make decisions about whether to trust its predictions or continue exploring. In our semantic map prediction problem, estimating the uncertainty over the semantic predictions at each location of the map enables understanding what the model does not know. We considered two types of uncertainty we face in modeling vision problems with deep learning: aleatoric and epistemic uncertainty (Kendall & Gal, 2017; Gal, 2016). First, aleatoric uncertainty is

the system uncertainty. Suppose the true probability of a sofa being at a specific location given an observation is 30%. Consider a scenario where our target is *sofa* and our model estimates the true probability of 30% that a sofa is at the specific location. Our model would be correct regardless of whether the sofa is present at that location. This uncertainty is reflected in the probability output of our model. We denote this model $f : (p_t, \hat{s}_t; \theta) \mapsto \hat{m}_t$ where $\theta$ are the parameters of $f$.

Second, epistemic uncertainty captures the uncertainty over the model's parameters. In training, our objective is to improve the prediction model by identifying cases it underperforms. We use epistemic uncertainty to formulate this objective, as samples with high epistemic uncertainty are associated with increased information gain. We recall that $f$ is a classifier trained with the cross-entropy loss, so the output of $f$ is a probability distribution. In order to estimate epistemic uncertainty, we consider the probabilistic interpretation $P(m_t|p_t, \hat{s}_t, \theta)$ of our model $f$ which defines a likelihood function over the parameters $\theta$. The parameters $\theta$ are random variables sampled from the distribution $q(\theta)$. We construct $f$ as an ensemble of two-stage segmentation models defined over the parameters $\{\theta_1, ..., \theta_N\}$. Variance between models in the ensemble comes from different random weight initializations in each network (Pathak et al., 2019; Sekar et al., 2020). Our model estimates the true probability distribution $P(m_t|p_t, \hat{s}_t)$ by averaging over sampled model weights, $P(m_t|p_t, \hat{s}_t) \approx \mathbb{E}_{\theta \sim q(\theta)} f(p_t, \hat{s}_t; \theta) \approx \frac{1}{N} \sum_{i=1}^N f(p_t, \hat{s}_t; \theta_i)$ (Lakshminarayanan et al., 2017; Gal et al., 2017). Then, following prior work (Seung et al., 1992; Pathak et al., 2019), the epistemic uncertainty can be approximated from the variance between the outputs of the models in the ensemble, $\text{Var} f(p_t, \hat{s}_t; \theta)$. We use uncertainty estimation in two distinct ways in our method. First, during training of the semantic predictor we actively select locations of the map with high information gain (Section 3.2.1). Second, during object-goal navigation we actively choose long-term goals that encourage the agent to explore in search of the target object (Section 3.3).

### 3.2.1 ACTIVE TRAINING

A typical procedure for training the semantic map predictor would be to sample observations along the shortest path between two randomly selected locations in a scene. However, this results in collecting a large amount of observations from repetitive or generally uninteresting areas (e.g. spaces devoid of objects of interest). We use that naive strategy for pre-training (around 900K examples), and formulate a following step where we actively collect training samples using an information gain objective. We choose destinations for which the predictions of our pre-trained models maximize this objective. Since we can interpret our hallucinated map $\hat{m}_t$ as a prediction $\{\hat{s}_{t+1}, ..., \hat{s}_{t+T}\}$ of the semantic segmentation of future observations over some window $T$, our greedy objective for information gain allows us to collect data in directions where we expect to make the most informative observations. The agent then moves using a local policy (described in Section 3.3.3). The models are then fine-tuned using the collected training examples (around 500K).

We evaluate which observations will be useful data for training by selecting $(x, y)$-grid locations to maximize an approximation of $I(m_t; \theta|p_t, \hat{s}_t)$. $I(m_t; \theta|p_t, \hat{s}_t)$ denotes the information gain with respect to the map $m_t$ from an update of the model parameters $\theta$ given the occupancy observation $p_t$ and semantic map crop $\hat{s}_t$. For brevity, we specify a grid location as $l_j \in \{l_1, ..., l_k\}$ where $k = hw$ for an $h \times w$ map region over which our model $f$ estimates $\hat{m}_t$. We select locations from the map which have the maximum epistemic uncertainty as a proxy for maximizing information gain (Pathak et al., 2019; Seung et al., 1992). To this end, we define the average epistemic uncertainty across all classes. We select locations $l_j$ from the map at time $t$ with the greedy policy

$$\arg \max_{l_j} I(m_t; \theta|p_t, \hat{s}_t) \approx \arg \max_{l_j} \frac{1}{|C^s|} \sum_{C^s} \text{Var} f(p_t, \hat{s}_t; \theta). \tag{1}$$

In practice, these locations are selected from the accumulated uncertainty estimates in the global map. Alternatives to our chosen active training strategy include policies maximizing entropy (Shannon, 1948) or an approximation of model information gain using predictive entropy (BALD) (Houlsby et al., 2011). We experimentally compare these alternatives to our approach in Section 4.1.

### 3.3 GOAL NAVIGATION POLICY

We study the problem of target-driven navigation within novel environments, which can be formulated as a partially observable Markov decision process (POMDP) $(S, A, O, P(s'|s, a), R(s, a))$. We are

interested in defining a policy that outputs goal locations as close as possible to a target class $c$. The state space $S$ consists of the agent's pose $x$ and the semantic predictions $\hat{m}_t$ accumulated over time in the global map. The action space $A$ is comprised of the discrete set of locations $h \times w$ over the map. The observation space $O$ are the RGB-D egocentric observations, and $P(s'|s, a)$ are the transition probabilities. A common reward choice for a supervised policy would be $R(s, a) = D(s, c) - D(s', c)$ which is the distance reduced between the agent and the target, where $D(., .)$ is distance on the shortest path. However, this results in a target-dependent learned policy, which would require re-training when a novel target is defined. Therefore, we formulate a policy which accumulates the predicted semantic crops $\hat{m}_t$ at every time-step, and leverages the predicted class probabilities along with uncertainty estimation over the map locations to select informative goals.

### 3.3.1 UPPER CONFIDENCE BOUND FOR GOAL SELECTION

We now use our uncertainty-driven approach to exploration to explicitly propose an objective for goal selection. During task execution at test time, $f$ cannot gain information because we do not update the model online with agent observations. However, the agent gains information by accumulating observations and successive predictions in the global map. We construct a policy in order to select goals from unobserved map locations using this accumulated information. The idea behind our policy is simple; if the agent is not very confident of where the target is situated, then it should prioritize exploration, otherwise it should focus more on exploiting its knowledge of candidate goal locations.

Since our task is target-driven, we can narrow our information gain objective to reduce uncertainty in areas of the map with the highest uncertainty about the target class. We denote $f_c$ as the function $f$ which only returns the values for a given target class $c$. For class $c$, our ensemble $f_c$ estimates $P_c(m_t|p_t, \hat{s}_t)$, the probability class $c$ is at location $i$ given an observation $p_t$ and semantic segmentation $\hat{s}_t$ for each map location. The target class uncertainty at test time is given by the variance over the target class predictions $\mathrm{Var} f_c(p_t, \hat{s}_t; \theta)$.

We propose selecting goals using the upper confidence bound of our estimate of the true probability $P_c(m_t|p_t, \hat{s}_t)$ in order to select locations with high payoffs but also high potential for our model to gain new information. Upper confidence bounds have long been used to balance exploration and exploitation in problems with planning under uncertainty (Auer et al., 2002; Azar et al., 2017; Chen et al., 2017). We denote $\sigma_c(p_t, \hat{s}_t) = \sqrt{\mathrm{Var} f_c(p_t, \hat{s}_t; \theta)}$ as the standard deviation of the target class probability, and we denote $\mu_c(p_t, \hat{s}_t) = \frac{1}{N} \sum_{i=1}^{N} f_c(p_t, \hat{s}_t; \theta_i)$. Then, we observe the upper bound $P_c(m_t|p_t, \hat{s}_t) \le \mu_c(p_t, \hat{s}_t) + \alpha_1 \sigma_c(p_t, \hat{s}_t)$ holds with some fixed but unknown probability where $\alpha_1$ is a constant hyperparameter. Following Chen et al. (2017), we use this upper bound and select goals from any map region the agent has observed or hallucinated with the policy

$$\arg\max_{l_j} \left( \mu_c(p_t, \hat{s}_t) + \alpha_1 \sigma_c(p_t, \hat{s}_t) \right). \tag{2}$$

In practice, this is evaluated over our predictions and uncertainty estimations accumulated over time.

### 3.3.2 ALTERNATIVE STRATEGIES

While we found our proposed upper bound strategy to choose the best candidate goal locations, we hypothesized several alternative policies which yield competitive performance. We consider the following interpretations of our uncertainty and probability estimates for object goal navigation.

- High $\mu_c(p_t, \hat{s}_t)$, Low $\sigma_c(p_t, \hat{s}_t)$ : we are fairly certain we found the target object
- High $\mu_c(p_t, \hat{s}_t)$, High $\sigma_c(p_t, \hat{s}_t)$ : we are uncertain we found the target object
- Low $\mu_c(p_t, \hat{s}_t)$, High $\sigma_c(p_t, \hat{s}_t)$ : we are uncertain we did *not* find the target object
- Low $\mu_c(p_t, \hat{s}_t)$, Low $\sigma_c(p_t, \hat{s}_t)$ : we are fairly certain we did *not* find the target object

From these interpretations we see that our upper bound strategy risks choosing locations with high potential information gain (high variance) over locations where we are fairly certain we found the target object (high probability, low variance). To consider other exploration and exploitation tradeoffs, we also observe the lower bound $P_c(m_t|p_t, \hat{s}_t) \ge \mu_c(p_t, \hat{s}_t) - \alpha_1 \sigma_c(p_t, \hat{s}_t)$ holds with some fixed but unknown probability. Then, we can formulate the following alternative goal selection strategies.

**Lower Bound Strategy.** Lower bound strategies optimize safety (Galichet et al., 2013). We can differentiate between multiple locations which have high probability of containing our target class by choos-

| Method | SPL (%) ↑ | Soft SPL (%) ↑ | Success (%) ↑ | DTS (m) ↓ |
|---|---|---|---|---|
| L2M-Entropy-UpperBound | $9.4 \pm 0.4$ | $15.5 \pm 0.4$ | $29.3 \pm 0.9$ | $3.666 \pm 0.074$ |
| L2M-Offline-UpperBound | $9.6 \pm 0.4$ | $16.0 \pm 0.4$ | $30.1 \pm 0.9$ | $3.528 \pm 0.070$ |
| L2M-BALD-UpperBound | $9.6 \pm 0.4$ | $15.7 \pm 0.4$ | $30.4 \pm 0.9$ | $3.664 \pm 0.074$ |
| L2M-Active-UpperBound | $\mathbf{13.3} \pm 0.5$ | $\mathbf{19.1} \pm 0.4$ | $\mathbf{34.3} \pm 0.9$ | $\mathbf{3.495} \pm 0.077$ |

Table 1: Comparison of different active training strategies on object-goal navigation.

ing the one with the lowest uncertainty with the objective: $\arg\max_{l_j} (\mu_c(p_t, \hat{s}_t) - \alpha_1 \sigma_c(p_t, \hat{s}_t))$. However, this strategy does not explore regions with high uncertainty to gain information.

**Mixed Strategy.** We can try to balance the pros and cons of the lower and upper bound strategies by switching between the two based on how high the probability is that we found the target object. We tune a hyperparameter $\alpha_2$ to determine the cutoffs for "high" and "low" values of $\mu_c(p_t, \hat{s}_t)$. We select goals with the objective: $\arg\max_{l_j} (\mu_c(p_t, \hat{s}_t) + \text{sgn}(\alpha_2 - \mu_c(p_t, \hat{s}_t))\alpha_1 \sigma_c(p_t, \hat{s}_t))$, so that we choose a safe strategy via our lower bounds when the probability of the class at a location is high and an exploration strategy via our upper bounds when the probability of the class is not high.

**Mean Strategy.** To evaluate whether our uncertainty estimate is useful, we also consider the following objective which does not incorporate uncertainty: $\arg\max_{l_j} \mu_c(p_t, \hat{s}_t)$.

### 3.3.3 LOCAL POLICY

Finally, in order to reach a selected goal in the map, we employ the off-the-shelf deep reinforcement learning model DD-PPO (Wijmans et al., 2019) without re-training. This model is trained for the task of point-goal navigation and at each time-step receives the egocentric depth observation and the current goal, and outputs the next navigation action for our agent.

## 4 EXPERIMENTS

We perform experiments on the Matterport3D (MP3D) (Chang et al., 2017) dataset using the Habitat (Savva et al., 2019) simulator. MP3D contains reconstructions of real indoor scenes with large appearance and layout variation, and Habitat provides continuous control for realistic agent movement. We use the standard train/val split as the test set is held-out for the online Habitat challenge, which contains 56 scenes for training and 11 for validation. We conduct three key experiments. First, we evaluate the performance of our semantic map predictor both in terms of navigation and map quality under different

| Method | IoU (%) | F1 (%) |
|---|---|---|
| L2M-Offline | 20.1 | 30.5 |
| L2M-Entropy | 20.7 | 31.2 |
| L2M-BALD | 21.2 | 31.8 |
| L2M-Active | **25.6** | **38.3** |

Table 2: Comparison of active training methods in semantic map prediction.

active training strategies (sec. 4.1). Second, we compare our method to other navigation strategies on reaching semantic targets and provide ablations over different goal-selection strategies (sec. 4.2). Finally, we conduct an error analysis over the effect of the stop decision and local policy in the overall performance (sec. 4.3). For all experiments we use an ensemble size $N = 4$. Trained models and code can be found here: `https://github.com/ggeorgak11/L2M`.

We follow the definition of the object-goal navigation task as described in Batra et al. (2020). Given a semantic target (e.g. chair) the objective is to navigate to any instance of the target in the scene. The agent is spawned at a random location in a novel scene which was not observed during the semantic map predictor training. The agent has access to RGB-D observations and pose provided by the simulator without noise. We note that estimating the pose from noisy sensor readings is out of the scope of this work and can be addressed by incorporating off-the-shelf visual odometry (Zhao et al., 2021). The action space consists of MOVE_FORWARD by $25cm$, TURN_LEFT and TURN_RIGHT by $10°$ and STOP. An episode is successful if the agent selects the STOP action within a certain distance ($1m$) from the target and must be completed within a specific time-budget (500 steps). Unless otherwise stated, for our experiments we use a representative set of 11 object goal categories present in MP3D: *chair*, *sofa*, *bed*, *cushion*, *counter*, *table*, *plant*, *toilet*, *tv*, *cabinet*, *fireplace* and generated 2480 test episodes across the validation scenes. To evaluate all methods we report the following metrics: (1) **Success:** percentage of successful episodes, (2) **SPL:** Success weighted by path length, (3) **Soft SPL:** Unlike SPL which is 0 for failed episodes, this metric is distance covered

| Method | SPL (%) ↑ | Soft SPL (%) ↑ | Success (%) ↑ | DTS (m) ↓ |
|---|---|---|---|---|
| Random Walk | $0.2 \pm 0.1$ | $1.0 \pm 0.2$ | $0.3 \pm 0.1$ | $5.408 \pm 0.141$ |
| Segm. + ANS [2] + OracleStop | $11.1 \pm 0.6$ | $12.2 \pm 0.4$ | $15.5 \pm 0.7$ | $4.982 \pm 0.087$ |
| L2M-Offline-FBE [1] | $5.1 \pm 0.3$ | $10.1 \pm 0.3$ | $20.6 \pm 0.8$ | $4.304 \pm 0.074$ |
| L2M-Offline-UpperBound | $9.6 \pm 0.4$ | $16.0 \pm 0.4$ | $30.1 \pm 0.9$ | $3.528 \pm 0.070$ |
| L2M-Active-Mean | $9.3 \pm 0.4$ | $15.7 \pm 0.4$ | $29.8 \pm 0.9$ | $3.618 \pm 0.073$ |
| L2M-Active-LowerBound | $9.6 \pm 0.4$ | $15.7 \pm 0.4$ | $30.4 \pm 0.9$ | $3.613 \pm 0.073$ |
| L2M-Active-Mixed | $10.4 \pm 0.4$ | $16.8 \pm 0.4$ | $30.8 \pm 0.9$ | $3.620 \pm 0.074$ |
| L2M-Active-UpperBound | $\mathbf{13.3} \pm 0.5$ | $\mathbf{19.1} \pm 0.4$ | $34.3 \pm 0.9$ | $\mathbf{3.495} \pm 0.077$ |
| SemExp [3] | $16.5 \pm 0.9$ | - | $28.1 \pm 1.2$ | $4.848 \pm 0.074$ |
| L2M-Active-UpperBound | $14.8 \pm 0.7$ | $20.0 \pm 0.6$ | $34.8 \pm 1.3$ | $3.669 \pm 0.114$ |

Table 3: Comparison against baselines and variations of our method. Baselines are adapted from [1] Yamauchi (1997), [2] Chaplot et al. (2020b), and [3] Chaplot et al. (2020a). Comparisons above the double horizontal line were carried out on our set of 11 objects while the comparison below the double line against SemExp was carried out on the 6 objects reported in [3]. Note that the discrepancy between the results of SemExp reported here and those in [3] are due to different sets of test episodes.

towards the goal weighted by path length. (4) **DTS:** geodesic distance of agent to the goal at the end of the episode. Different variants of our method are evaluated in the following experiments. We follow the naming convention of *L2M-X-Y* where *X*, *Y* correspond to the map predictor training strategy and the goal-selection strategy respectively. For example, *L2M-Active-UpperBound* refers to our proposed method that uses the semantic map predictor after active training (Section 3.2.1) and Eq. 2 for goal selection. More details of specific variants are provided in the following sections.

## 4.1 EVALUATION OVER ACTIVE TRAINING METHODS

We evaluate the impact of our approach to actively training our semantic mapping predictor by comparing different ensemble-based active training strategies over the quality of the predicted maps and object-goal navigation. The semantic map prediction is evaluated on the popular segmentation metrics of Intersection over Union (IoU), and F1 score. We use nine classes: *unknown*, *floor*, *wall*, *bed*, *chair*, *cushion*, *sofa*, *counter*, *table*. 17900 test examples were collected in the 11 validation scenes which had not been observed during training. The evaluation is conducted on predicted map crops of size $64 \times 64$ that are collected in sequences of length 10. The purpose of this experiment is to ascertain the quality of the predicted mapped area around the agent while it is performing a navigation task. Results are shown in Table 1 (object-goal navigation) and Table 2 (map prediction). The variant *Offline* is our semantic map prediction model without fine-tuning with an active strategy, *BALD* adapts the BALD objective (Houlsby et al., 2011) for actively training our ensemble, and *Entropy* refers to our model fine-tuned with an entropy objective for active training (Shannon, 1948). For the navigation comparison all methods use the upper bound objective from Eq. 2. While the baselines report similar performance to each other, our method gains $4.4\%$ in IoU, $6.5\%$ in F1, $3.9\%$ in Success, and $3.7\%$ in SPL from the second best method. This indicates that *L2M-Active* is more effective in targeting data with high epistemic uncertainty during training

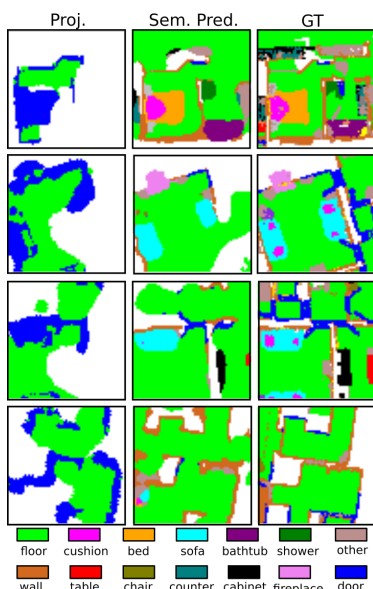

Figure 3: Qualitative map prediction results.

and validates our choice for the information gain objective presented in Section 4.1 of the main paper. Figure 3 shows qualitative results of *L2M-Active*.

## 4.2 COMPARISONS TO OTHER NAVIGATION METHODS

Here we evaluate *L2M* against three competitive baselines:

**L2M-Offline-FBE:** We combine the classical Frontier-based exploration (FBE) from Yamauchi (1997) for goal selection with our map predictor to facilitate the stop decision.

| Method | SPL (%) ↑ | Soft SPL (%) ↑ | Success (%) ↑ | DTS (m) ↓ |
|---|---|---|---|---|
| L2M | $12.5 \pm 0.7$ | $21.1 \pm 0.7$ | $32.7 \pm 1.6$ | $3.898 \pm 0.148$ |
| L2M + GtPath | $17.7 \pm 0.8$ | $20.6 \pm 0.8$ | $50.4 \pm 1.7$ | $3.545 \pm 0.175$ |
| L2M + OracleStop | $33.1 \pm 1.3$ | $31.5 \pm 0.9$ | $50.8 \pm 1.7$ | $3.585 \pm 0.134$ |
| L2M + GtPath + OracleStop | $52.3 \pm 1.3$ | $41.6 \pm 1.0$ | $80.5 \pm 1.4$ | $2.620 \pm 0.150$ |

Table 4: Ablations of our method that investigate the effect of stop decision and local policy.

**Segm+ANS+OracleStop:** This baseline uses Active Neural SLAM (ANS) as an exploration policy to traverse the map and our image semantic segmentation to detect objects. If the target object is detected, the agent navigates to that goal, and an oracle decides to stop the episode if the agent reaches the correct target. The agent does not have access to a semantic map.
**SemExp:** The method proposed in Chaplot et al. (2020a) which was the winner of the CVPR 2020 Habitat ObjectNav Challenge. Since the model that was used in the Habitat challenge is not publicly available, we compare against the six object categories reported in Chaplot et al. (2020a) using the variant of this method that used Mask R-CNN. Furthermore, since the MP3D evaluation episodes used in Chaplot et al. (2020a) are also not available, we ran SemExp on our set of evaluation episodes.

In addition, we evaluate variations of our method (*LowerBound*, *Mixed*, and *Mean*) as defined in section 3.3.2 with $\alpha_1 = 0.1$ and $\alpha_2 = 0.75$. Our results are presented in Table 3. We observe that our *L2M-Active-UpperBound* method outperformed all baselines in terms of success rate by a significant margin and is comparable to *SemExp* in terms of SPL. This result is not surprising since our upper bound strategy often chooses goals that maximize information gain over the map for effective exploration rather than choosing the shortest path to the goal. Interestingly, *L2M-Offline-FBE* outperforms *Segm.+ANS+OracleStop* even though the latter has access to the stop oracle (which results in a high SPL performance). This demonstrates the advantage of having access to our map prediction module for the object-goal navigation task. Furthermore, any performance gains of our method towards *L2M-Offline-FBE* are a direct result of our goal selection strategies. Regarding our L2M variations, the upper bound strategy performed best across all metrics. We note that we expect the mixed and upper bound strategies to have close performance results since by definition the mixed strategy executes the upper bound strategy whenever the probability of the target class at a given location is less than $\alpha_2 = 0.75$.

### 4.3 ERROR ANALYSIS

A common source of failure in navigation tasks is deciding to stop outside the success radius of the target. In this last experiment we investigate the effect of the stop decision in failure cases of our model by defining an oracle that provides our model with the stop decision within success distance of the target (*OracleStop*). In addition, we explore the contribution of the local policy in failure cases by replacing it with shortest paths estimated by the habitat simulator towards our selected goals (*GtPath*). The rest of the components follow our proposed *L2M-Active-UpperBound*. The evaluation for this experiment was carried out on a subset of 795 test episodes which are harder due to larger geodesic to euclidean distance ratio between the start position and the target and have larger mean geodesic distance than the rest of our test episodes. Table 4 illustrates our findings. We observe a significant increase of performance for all baselines. In the case of *L2M + GtPath* the performance gap suggests that the local policy has difficulties reaching the goals, while in the case of *L2M + OracleSTOP* it suggests that our model selects well positioned goals but our stop decision criteria fail to recognize a goal state. Finally, *L2M + GtPath + OracleSTOP* achieves mean success rate of $80\%$, thus advising further investigation of these components of our pipeline.

## 5 CONCLUSION

We presented *Learning to Map* (L2M), a novel framework for object-goal navigation that leverages semantic predictions in unobserved areas of the map to define uncertainty-based objectives for goal selection. In addition, the uncertainty of the model is used as an information gain objective to actively sample data and train the semantic predictor. We investigated different information gain objectives and found epistemic uncertainty to be the most effective for this problem. Furthermore, we proposed multiple goal-selection strategies and observed that balancing exploration and exploitation using upper confidence bounds of our predictions produced higher performance. Finally, our method outperformed competitive baselines on the Matterport3D dataset for object-goal navigation.

**Ethics Statement.** Our work advances the capability of autonomous robots to navigate in novel environments which can help create a positive social impact through technologies such as robot caregivers for medically underserved populations. However, our approach has technical limitations which can yield negative social consequences. Our semantic hallucination method does not model out-of-distribution scenes and is trained on data from homes in North America and Europe. If utilized for safety critical tasks such as medical caregiving in hospitals instead of homes or in homes in different regions of the world, our method would act on biases driven by home structure in the training set with potentially harmful outcomes. Another technical limitation of our work is our inability to model 3D relationships. We ground project 3D information from depth to a 2D map representation for occupancy, thus losing 3D spatial context. This can be important for objects such as cushions which are often on top of other objects such as sofas. Losing this context potentially contributes to our lower success rate on specific objects.

**Reproducibility Statement.** In Section 4, we include a footnote after the first sentence of the section with a link to our GitHub repository. This repository includes the code for our models and instructions for reproducing our results. We provide both a Docker image with the dependencies for running our code and instructions to install the required dependencies without using Docker. We give instructions for generating initial training data from the Habitat simulator as well as instructions for collecting training data using our active policy. We have instructions for training and testing all of the model variants described in our work. We provide Google Drive links to the test episodes we used to evaluate our models as well as the MP3D scene point clouds which include the semantic category labels for the 3D point clouds. Each of our trained models is also shared via Google Drive links, and we link to the pre-trained DD-PPO model provided by the authors of that work which we leverage in our experiments. In addition, implementation details can be found in section A.1 of the Appendix where we describe the hyperparameter values used during our experiments, the training procedure that we followed for the semantic map predictors, and we provide the pseudo-code algorithm for the execution of our method during a navigation episode.

ACKNOWLEDGMENTS

We would like to thank Samuel Xu for implementation support in evaluating baseline methods. Research was sponsored by the Honda Research Institute through the Curious Minded Machines project, by the Army Research Office under Grant Number W911NF-20-1-0080 and by the National Science Foundation under grants CPS 2038873, NSF IIS 1703319.

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

## A   APPENDIX

Here we provide the following additional material:

1. Implementation details.
2. Additional experimental results for semantic map prediction.
3. Per-object navigation results for the different active training strategies.
4. Per-object error analysis over stop decision and local policy.
5. Evaluation of our method on easy vs hard episodes.
6. Additional visualizations of semantic maps and navigation examples.

### A.1   IMPLEMENTATION DETAILS

---

**Algorithm 1:** L2M for ObjectNav

---

**Input:** Semantic target $c$;
Time $t = 0$ with current position $x_t$;
Stop decision probability threshold $S_p$;
Stop decision distance threshold $S_d$;
Replan interval $R$;
**while** $t <$ *max_steps_per_episode* **do**
  Observe RGB $I_t$, occupancy $p_t$;
  Segment $I_t$ and ground project to
    compute $\hat{s}_t$;
  Hallucinate semantic map
    $\hat{m}_{t,c} = \mu_c(p_t, \hat{s}_t)$;
  Estimate uncertainty;
  **if** *(Goal is reached) or*
  *(t mod R == 0)* **then**
    | Compute Eq. 2 to select goal $l^*$;
  **end**
  **if** $\hat{m}_{t,c} > S_p$ *at $l_j$ on path to $l^*$* **then**
    **if** $D(x_t, l_j) < S_d$ **then**
      | Stop decision;
    **end**
  **end**
  Navigate toward goal with DD-PPO;
  $t = t + 1$;
**end**

---

All UNets Ronneberger et al. (2015) in our implementation are combined with a backbone ResNet18 He et al. (2016) for providing initial encodings of the inputs. Each UNet has four encoder and four decoder convolutional blocks with skip connections. The models are trained in the PyTorch Paszke et al. (2017) framework with Adam optimizer and a learning rate of 0.0002. All experiments are conducted with an ensemble size $N = 4$. For the semantic map prediction we receive RGB and depth observations of size $256 \times 256$ and define crop and global map dimensions as $h = w = 64$, $H = W = 384$ respectively. We use $C^s = 27$ semantic classes that we selected from the original 40 categories of MP3D Chang et al. (2017) We generate the ground-truth for semantic crops using the 3D point clouds of the scenes which contain semantic labels. We executed training and testing on our internal cluster on RTX 2080 Ti GPUs. To train our final model, our image segmentation network first trained for 24 hours. Then, each model in our offline ensemble trained for 72 hours on separate GPUs. Finally, each model was fine-tuned on the actively collected data for 24 hours on separate GPUs.

Regarding navigation, we use a threshold of 0.75 on the prediction probabilities to determine the occurrence of the target object in the map, and a stop decision distance of $0.5m$. Finally, we re-select a goal every 20 steps. Algorithm 1 shows additional details regarding the execution of our method during an object-goal navigation episode.

### A.2   SEMANTIC MAP PREDICTION

Here we provide additional results for our semantic map predictor, including evaluation over occupancy predictions (*unknown*, *occupied*, *free*). The set-up for this experiment is the same as in Section 4.1 of the main paper. The purpose of this evaluation is to demonstrate the superiority of our method to possible non-prediction alternatives such as using directly the projected depth, or ground-projected image segmentation. To this end we compare against the following baselines:

- **Depth Projection:** Occupancy map estimated from a single depth observation.
- **Multi-view Depth Projection:** Occupancy map accumulated over multiple views of depth observations.
- **Image Segmentation Projection:** Our semantic segmentation model that operates on image observations, followed by projection of resulting labels.

| Occupancy Prediction | | | |
| --- | --- | --- | --- |
| Method | Acc (%) | IoU (%) | F1 (%) |
| Depth Proj. | 31.2 | 14.5 | 24.3 |
| Multi-view Depth Proj. | 48.5 | 31.0 | 47.0 |
| L2M-Offline | 65.2 | 45.5 | 61.9 |
| L2M-Active | **68.1** | **48.8** | **65.0** |
| Semantic Prediction | | | |
| Image Segm. Proj. | 13.9 | 5.9 | 10.2 |
| Sem. Sensor Proj. | 16.9 | 9.1 | 16.0 |
| Multi-view Sem. Sensor Proj. | 27.6 | 18.9 | 31.4 |
| L2M-Offline | 31.2 | 20.1 | 30.5 |
| L2M-Active | **33.5** | **25.6** | **38.3** |

Table 5: Comparison of our occupancy and semantic map predictions to non-prediction alternatives.

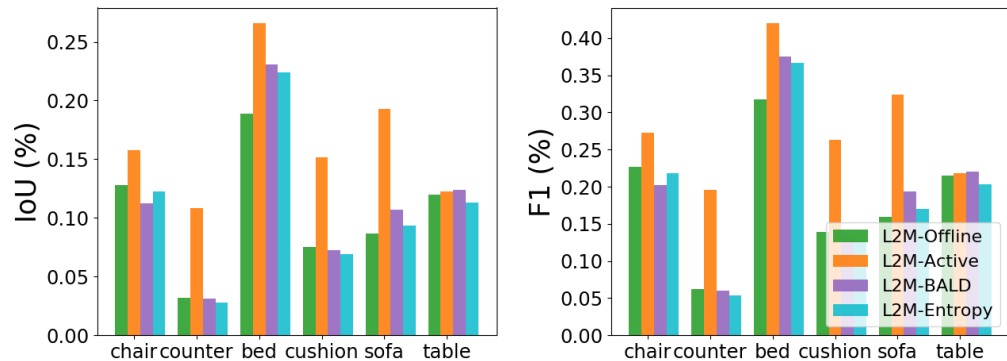

Figure 4: Semantic mapping results per class over different active training methods.

- **Semantic Sensor Projection:** Semantic map generated by single-view ground-truth semantic sensor images provided by the simulator.

- **Multi-view Semantic Sensor Projection:** Same as the previous baseline, but with multiple views accumulated in the semantic map.

We present mean values for accuracy, intersection over union (IoU), and F1 score in Table 5. Both our approaches outperform all baselines by a significant margin. This suggests that our predictions of unobserved areas can provide more useful information to the agent than only relying on accumulated views from egomotion. In the case of semantic prediction our results are even more compelling as they are compared against the ground-truth semantic sensor of the Habitat simulator. Note that the *Multi-view Sensor* baseline gets far from perfect score for two reasons: 1) it still contains unobserved areas, and 2) due to the pooling of the labels in the lower spatial dimensions of the top-down map (which affects all projections). In contrast, our approach is unaffected by this problem as we learn to predict semantics from depth projected inputs.

### A.3   PER OBJECT EVALUATION FOR ACTIVE TRAINING METHODS

In this section we show per object results over map prediction quality and object-goal navigation for the different active training strategies (*L2M-BALD*, *L2M-Entropy*, *L2M-Offline*) introduced in Section 4.1 of the main paper against our proposed *L2M-Active*. We show results over a representative subset of six objects *chair*, *bed*, *cushion*, *counter*, *sofa*, *table* in Figure 4 (map prediction) and Figure 5 (object-goal navigation). In both cases we observe that the performance gap between *L2M-Active* and the baselines is larger on more challenging target classes such as cushion and counter. This suggests that our active training successfully selected examples with high information value as opposed to *L2M-BALD* and *L2M-Entropy* which do not show significant improvement over *L2M-Offline*.

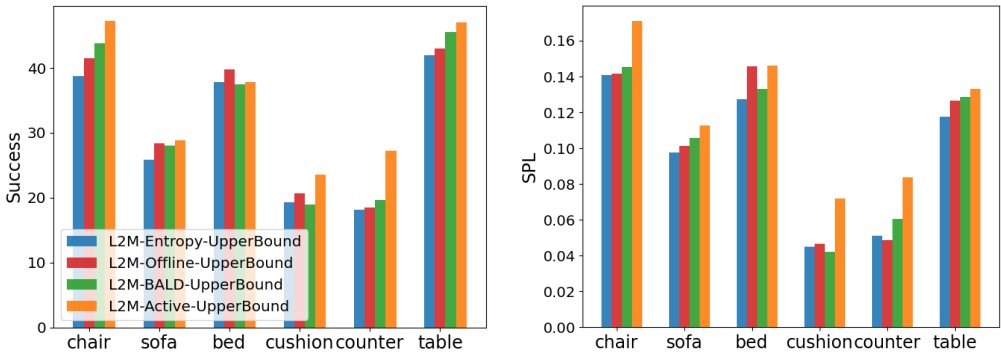

Figure 5: Navigation results per class over different active training methods.

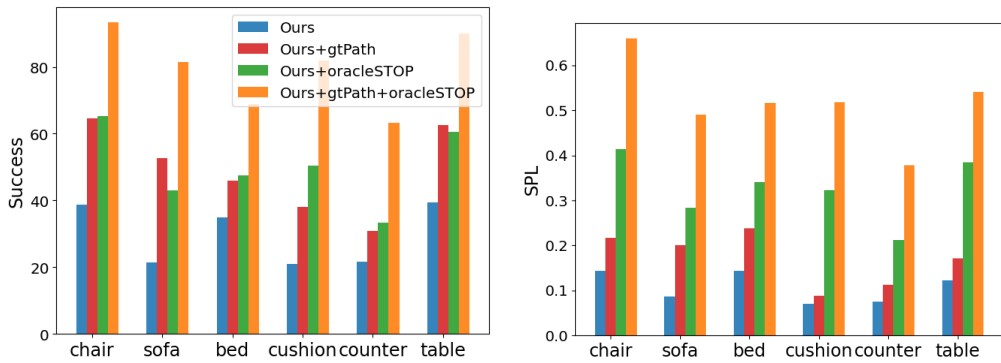

Figure 6: Ablations of our method that investigate the effect of the stop decision and local policy.

## A.4 PER OBJECT ERROR ANALYSIS

Here we present additional results to the experiment in section 4.3 where we investigated the effect of the stop decision and local policy in failure cases of our model. To do so, we combined our *L2M-Active* method with an oracle that stops the agent within success distance of the target (*OracleStop*) and replaced the local policy with shortest paths estimated by the habitat simulator to reach our selected goals (*GtPath*). Results are shown in Figure 6. The largest performance improvement (especially for SPL) is seen when the *OracleStop* is enabled, as opposed to *GtPath*, suggesting that a very common failure case is recognizing that we've reached the target. Perhaps unsurprisingly, this is also more pronounced in categories where our map predictor seems to be underperforming (see Figure 4) such as cushion, counter and table. This result also indicates that our method selects well positioned goals across all object categories, but we often fail to either reach the target due to errors in the local policy or fail to recognize a goal state.

## A.5 EASY VS HARD EPISODES

Another way of evaluating the impact of our proposed method is by analyzing its performance with respect to *easy* and *hard* episodes. We generated 1685 *easy* and 795 *hard* episodes, which combined make up our entire test set used in Section 4 of the main paper. The *hard* episodes are generated with larger geodesic to euclidean distance ratio between the starting position and the target ($1.1$ vs $1.05$), which translates to more obstacles present in the path, and have larger mean geodesic distance ($6.5m$ vs $4.5m$). The results are visualized in Figure 7. It is worth noting that the performance gap is higher in the case of *hard*. Our actively trained semantic mapping model seeks out difficult data during training, resulting in more consistently high performance on both *easy* and *hard* episodes than the models trained only with offline data (*L2M-Offline*), or using different information-gain objectives (*L2M-Entropy*, *L2M-BALD*).

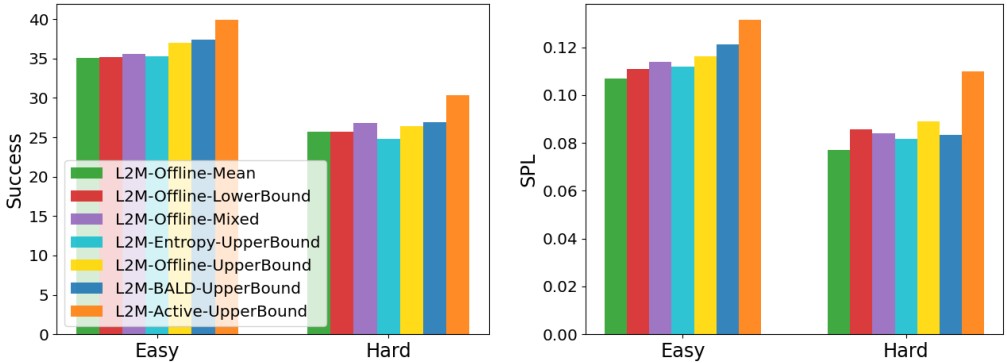

Figure 7: Our actively trained semantic mapping model shows consistently higher performance on hard episodes.

## A.6 ADDITIONAL VISUALIZATIONS

Finally, we provide some additional visualizations. Example navigation episodes are shown in Figure 8. A set of semantic predictions are shown in Figure 9, while Figure 10 showcases predictions from the individual models in the ensemble, qualitatively demonstrating the variation within the ensemble over semantic predictions.

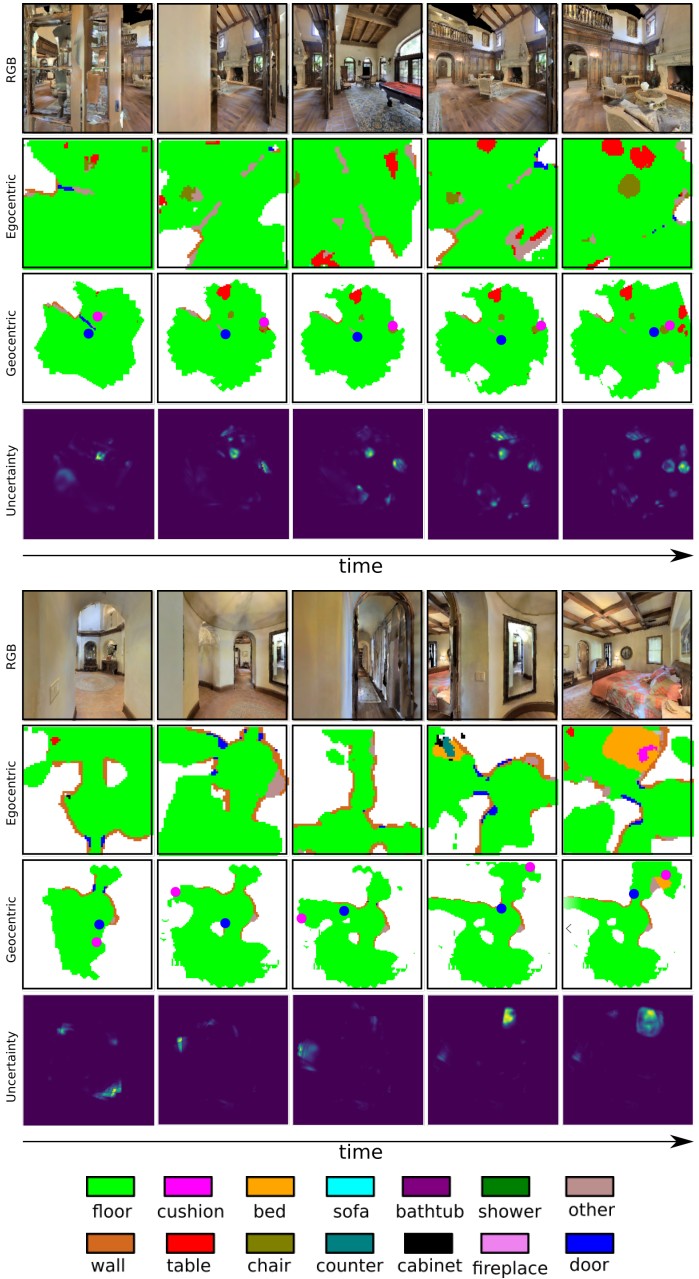

Figure 8: Examples of successful navigation episodes where the targets are "chair" (top) and "bed" (bottom). For each example, first row shows egocentric RGB observations, second row are the egocentric semantic predictions, third row are the registered geocentric predictions, and the last row shows the model uncertainty of the target class over the geocentric predictions (brighter color signifies higher uncertainty). The agent is shown as a blue dot, and the current goal as magenta.

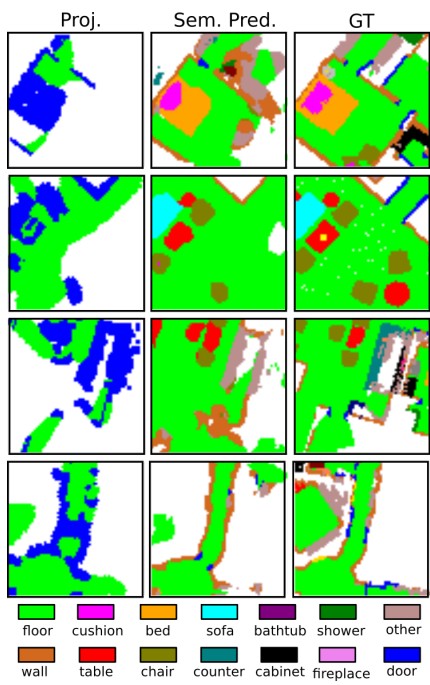

Figure 9: Qualitative semantic prediction results using our *L2M-Active* approach.

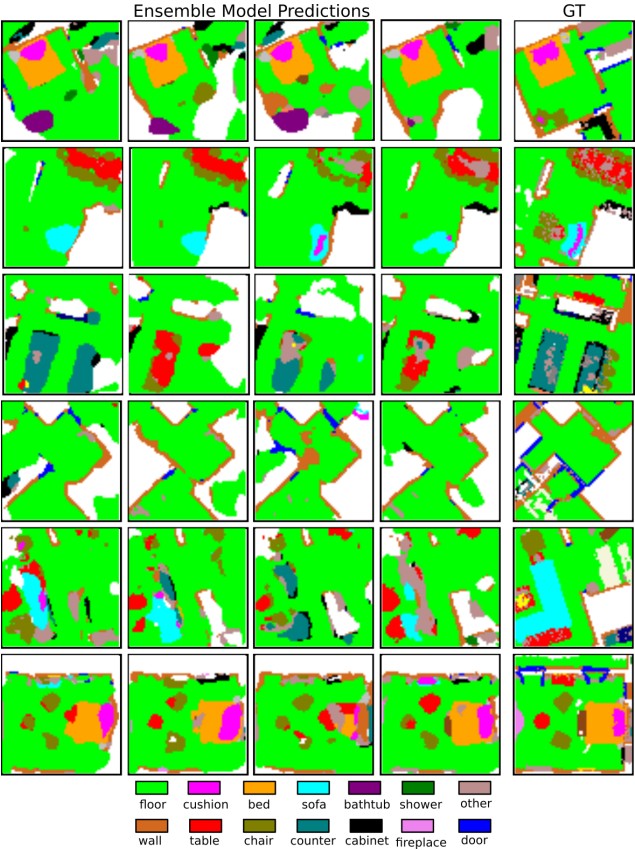

Figure 10: Qualitative semantic prediction from the individual models in the ensemble (first four columns) using our *L2M-Active* approach.

