# OpenReview forum: "Learning to Map for Active Semantic Goal Navigation"
_ICLR.cc/2022/Conference — ICLR 2022 Poster_

### Official Review · Reviewer_qy8x · 2021-11-04

**Correctness:** 4
**Technical Novelty And Significance:** 3
**Empirical Novelty And Significance:** 2
**Recommendation:** 6
**Confidence:** 4

**Main Review:**

The paper explains the high level ideas fairly clearly. However, the proposed method has a few components. It would be easier to follow the details if there are more structured algorithm boxes that go with components in section 3. For instance, it is not very clear what the training loss for the segmentation model is does it account for class imbalance etc. Some of the details are present in the appendix but need a bit of hopping around to figure out.

The motivation and high-level idea makes sense. Leveraging priors on typical semantic arrangement of objects for exploration seems natural. That said leveraging these priors as proposed in the paper requires very precise semantic segmentation maps which are fairly expensive to obtain. It seems like the proposed method in the paper can be applied to bounding boxes which are much cheaper to obtain as opposed to semantic segmentation.

The uncertainty estimation relies on an ensemble of models trained for the semantic map prediction. This can be fairly expensive especially with semantic segmentation models. There are other strategies to estimate uncertainty using a single model (https://arxiv.org/pdf/1506.02142.pdf). Have the authors considered these alternatives?

The experimental evaluation focuses on comparison with baselines from chaplot et.al where the problem setup and assumptions are a bit different. However, the overall pipeline seems fairly similar and leverages similar high-level ideas (https://arxiv.org/pdf/2006.09367.pdf). It is unclear what the main distinctions from this line work the current paper is contributing. Moreover, semantic segmentation is typically less robust that instance segmentation and tends to ignore smaller objects. In the experimental results is there a breakdown of navigation queries to semantic categories which are small and are only represented by a small fraction of the training data.

A simple baseline that seems to be missing is just using the projected semantic segmentation map without trying to guess the semantic layout in unseen regions to perform the navigation tasks. Are the navigation queries such that they cannot be solved with just what is observed is that ensured in the experiments?

**Summary Of The Paper:**

The paper proposes a method for leveraging semantic predictions in unobserved areas to help agent navigation. The strategy is motivated by how humans leverage priors on house layouts to perform navigation and tasks without being the house before. The key idea is to train a semantic segmentation model which can predict plausible class assignments in unseen regions. Then use the uncertainty estimates during navigation to pick goals when trying to reach a particular target category.

**Summary Of The Review:**

The high-level ideas the paper is based seem natural and interesting. However, fairly similar ideas have been proposed in prior work which is cited by the paper. It seems to me that the distinctions are mostly in the details a lot of which emerge from the slight differences in the problem setup.

Post-discussion: The authors have addressed the main concerns about distinctions from prior work. Therefore, I am improving my rating.

---

> ### Author Response · Authors · 2021-11-18
> **Reply to reviewer qy8x - Part 1**
>
> Q: For instance, it is not very clear what the training loss for the segmentation model is does it account for class imbalance etc. Some of the details are present in the appendix but need a bit of hopping around to figure out.
>
> A: We kindly ask the review to clarify their comment regarding how the clarity of the paper could be improved. In particular, the subsections in Section 3 directly correspond to algorithm boxes in Figure 1. The Semantic Map Prediction box is described in Section 3.1. The Goal Selection Policy box is described by Section 3.2 during training and Section 3.3 during testing. The Local Policy box is described by Section 3.3.3.
>
> The training loss for the segmentation model is pixel wise cross entropy for occupancy and semantic labels as described in Section 3.1. We tried weighting the results for class imbalance based on the support of different objects in the dataset, and it did not improve performance.
>
> Q: It seems like the proposed method in the paper can be applied to bounding boxes which are much cheaper to obtain as opposed to semantic segmentation.
>
> A:Our motivation for semantic segmentation is to model both “things” (tables, chairs etc) and “stuff” (floor, walls). Our intuition is that this is a stronger approach than using bounding boxes for predicting unseen map regions because it enables learning the holistic layout priors that include the scene structure. Though we believe semantic segmentation is the correct modeling approach, we agree that segmentation maps can be expensive to obtain. However, some recent work on automatic semantic labeling could help alleviate this problem (https://3dscenegraph.stanford.edu/).
>
> Q: There are other strategies to estimate uncertainty using a single model (https://arxiv.org/pdf/1506.02142.pdf). Have the authors considered these alternatives?
>
> We did consider using a Bayesian neural network approach but decided to pursue the ensemble-based methods due to the arguments from Lakshiminarayanan (https://arxiv.org/pdf/1612.01474.pdf) which we found pertinent to our problem. We kindly refer the reviewer to our response to the first question from reviewer Nmyk.
>
> Q:  It is unclear what the main distinctions from this line work the current paper is contributing.
>
> A: We kindly ask the reviewer to clarify their statement that the problem setup and assumptions between our work and Chaplot et al. (2020a) are different. We believe that the problem we solve is identical. We both define methods addressing the ObjectNav task per the exact definition in Batra et al. (2020).
>
> With respect to the methodology, we believe that our approach is significantly distinct from the SemExp approach by Chaplot et al. (2020a). In both SemExp and our work, goals are selected and passed to a local policy, but the goal selection policy is very different in each case. The SemExp approach uses ground projected MaskRCNN instance segmentations of observed data only as input to a trained reinforcement learning policy which directly predicts goal locations. In our work, we hallucinate unseen regions of a semantic map and select goals directly from this map with an objective function which does not require target specific training.
>
> Our approach also enables successful navigation in scenarios where the agent faces high uncertainty. If the agent using SemExp faces an unfamiliar environment, the navigation policy makes decisions with incorrect biases. Due to our uncertainty driven objective, in our model, if the agent faces an unfamiliar environment, they navigate based on uncertainty to gain information and ultimately navigate more successfully.
>
> The high level components of SemExp and our method may at first glance seem similar since we address the same problem, and methods performing object goal navigation all need to both process their visual observations and use them in a policy which selects actions to reach navigation goals. However, as discussed, the content of most of these components is unique for our method and SemExp, so methodologically, the approaches are quite different. Beyond a direct performance comparison, our uncertainty-driven navigation policy which does not require target specific training is a distinct contribution unaddressed by SemExp.

---

> > ### Comment · Reviewer_qy8x · 2021-11-27
> > **Thank you for all the detailed comments.**
> >
> > The comments address a large fraction of my concerns. Thanks for the detailed explanations and the additional experiments. I would suggest including some of the discussion here about Chaplot et al. (2020a) briefly in the related work.

---

> > > ### Author Response · Authors · 2021-11-29
> > > **Response to "Thank you for all the detailed comments."**
> > >
> > > Thank you for the suggestion! We will include the additional discussion in our related work.
> > >
> > > Since our response addressed a large number of your concerns, would you be willing to increase your rating of our paper? Specifically, we hope we have increased your confidence in the novelty of our navigation pipeline which introduces both a semantic map prediction model and uncertainty-driven goal selection policies unique to our work.

---

> ### Author Response · Authors · 2021-11-18
> **Reply to reviewer qy8x - Part 2**
>
> Q: In the experimental results is there a breakdown of navigation queries to semantic categories which are small and are only represented by a small fraction of the training data.
>
> A: We discuss earlier in this response our reasons for performing semantic segmentation over instance segmentation. With respect to class imbalance, we also mention earlier in this response that we tried weighting results based on frequency of object representation in the training data and did not see significant improvement in the semantic hallucination performance.
>
> We did observe lower navigation performance on small objects. We showed a partial per object breakdown of both navigation and segmentation results for a subset of objects in Figures 4 and 5 in the supplemental material. After submission, we trained a new map predictor model using a higher resolution of 192x192 grid with 0.05m cells as opposed to our original 64x64 grid with 0.1m cells. We saw significant performance improvement particularly on small objects from increasing the map resolution. We provide a link to a plot of the semantic segmentation results https://drive.google.com/file/d/1d6GMNHjnKzWmlusLHm1-Xb6mQvDioYMS/view?usp=sharing.
>
> We have yet to complete evaluation of the navigation results with this improved map, but we expect these results to correspond to significant navigation improvements as well given the past relationship between the two in our work.
>
> Q: A simple baseline that seems to be missing is just using the projected semantic segmentation map without trying to guess the semantic layout in unseen regions to perform the navigation tasks. Are the navigation queries such that they cannot be solved with just what is observed is that ensured in the experiments?
>
> A: We believe the baseline the reviewer requests is given by Segm. + ANS + OracleStop. We note that using a segmentation map without hallucination requires a different goal selection policy since our policy relies on navigating outside of our current field of view using goal selection from a hallucinated map. In our baseline Segm. + ANS + OracleStop, ANS (Active Neural SLAM) is used as an exploration policy to traverse the map. Then, a semantic object detector is used on the agent observations. If the target object is detected, the agent navigates to that goal, and an oracle decides to stop the episode if the agent reaches the correct target. We note that the use of the semantic object detector is equivalent to selecting the goals from a semantic map composed only of observations since either the agent will think the target exists in an observation or not. At a later point in time, that labeling decision will be the same, so a goal selection policy based on those decisions would never go back to that location.
>
> We see that Segm. + ANS + OracleStop performs significantly worse on success rate than  L2M, though it achieves the third highest SPL in Table 3. Much of that performance result can be explained by the advantage we give this baseline with the OracleStop decision. In Table 4, we demonstrate that using an oracle to stop when we are within the success distance of an object adds approximately 20% to the SPL and success rate of L2M.
>
> In most navigation queries the target object is not within the field-of-view or within the immediate area around the agent. Therefore, navigation decisions need to be made either through map exploration or a navigation policy which implicitly or explicitly extrapolates map information outside of the field of view of the agent. Segm. + ANS + OracleStop performs map exploration and significantly underperforms. SemExp implicitly extrapolates map information outside of the field of view of the agent through a trained navigation policy. We explicitly predict the unseen map. In conclusion, making predictions explicitly (L2M) or implicitly (SemExp) regarding the unseen map regions has shown to outperform exploration on this task.

---

### Official Review · Reviewer_CF2f · 2021-11-04

**Correctness:** 3
**Technical Novelty And Significance:** 4
**Empirical Novelty And Significance:** 4
**Recommendation:** 8
**Confidence:** 4

**Main Review:**

Firstly, thanks to the authors for their efforts. This is a very well written and high quality paper and I absolutely enjoyed reading it! In my opinion, the paper scores high on novelty, provides an interesting extension to the status quo on how mappers are generally used by embodied agents and nicely extends ideas from prior work in order to incorporate them into the task at hand. More concretely, I believe the following contributions are particularly the strengths of this work:
+ An active learning formulation for training the mapper by selection of informative waypoints where the mapper is expected to gain maximal amount of information
+ Using the variance of an ensemble of mappers as a proxy for taking into account the current mapper’s uncertainty for selecting goal targets in the map constructed so far: the formulation is elegant and nice leads to a controllable exploitation-exploration trade-off.

The insights presented in the paper are intuitive (learning from maximally informative regions is obviously a more efficient strategy than randomly sampling training trajectories for the mapper), the formulation is elegant (I liked how the same ideas related to variance of predictions applied to both mapper training and goal selection) and overall seem potentially useful for the Embodied AI community.

The paper also contains experiments pertaining to all the logical design choices for their approach and the empirical results largely support the claims made by the authors (refer to the question/suggestions section for some exceptions to this).

The only major complaint I have with the paper is the absence of a thorough comparison with state-of-the-art. The authors do compare with a couple of related contemporary approaches in Tab. 3. However, ObjectNav on MP3D/Habitat is a well established task with an active leaderboard (https://eval.ai/web/challenges/challenge-page/580/leaderboard/1634) and a comparison with the best performing models from the leaderboard seem to be missing.

I also had minor issues with some missing bits of information pertaining to how the overall system is working. I encourage the authors to clarify these points in the discussion phase. For instance,
+ There seem to be a couple of different, but related decision making processes going on: (a) the agent has to select informative waypoints to optimally train the semantic map prediction module and (b) the agent has to select the target semantic goal location on the map built so far while being mindful of the uncertainties of the current semantic predictions. Although both these decisions are made using the same fundamental principles at the core, it is not clear to me the order in which these decisions are made. Does the agent alternate between the two? If I understand correctly, both cannot be happening at the same time. Once either of the modules selects a waypoint, the agent has to move towards the same. Some clarification on how the overall system works would be helpful in this context.
+ The authors mention that the optimizations (eqn. 1-4), in practice, happen over the geo-centric map. Does it happen only for the set of map locations observed so far? That is, are there any constraints on the l_j’s in the geo-centric map?
+ It is not immediately clear what is causing the variance in predictions for the ensemble of mappers. Is it because of different initializations? Or, is the training for each being warm-started on a different subset of the dataset?

Given below are some minor questions/points of clarifications:
+ From the description in Sec. 3.3.2, the theory seems to suggest that using the LowerBound strategy for goal selection is more conservative towards exploration, as compared to the UpperBound strategy. One would expect a higher SPL as a consequence of that (the map predictor training strategy staying the same across both). Why isn’t that the case in Tab. 3 (L2M-Active-LowerBound=9.6 v/s L2M-Active-UpperBound=13.3. SPL)? Also related to this, the “LowerBound" numbers for “Offline" map predictor training is missing (L2M-Offline-LowerBound)
+ There are two rows for the same experiment in Tab.3 (L2M-Active-UpperBound). I suspect the reason is the special case for the SemExp baseline where the comparison happens only for a subset of 6 objects categories. Would be better to mention this explicitly in the text / table caption.
+ This is purely speculative from the reviewer (doesn't factor into the review), but I was wondering if the same ideas would work in the space of features, as opposed to logits, as in current work (that are being projected onto the top-down map). Prior work (https://arxiv.org/abs/2010.01191) has shown that projection of features, rather than projected labels from egocentric segmentations circumvents the "label splatter" issue. I'm curious to know the authors' thoughts on this -- do they think that the same framework on variance of ensembles would work equally well in the semantic feature space?

**Summary Of The Paper:**

The paper presents a novel framework that learns how to construct spatial + semantic maps for target-driven semantic (ObjectGoal) navigation. The scene maps being learned can also hallucinate beyond the immediate observed regions by exploiting semantic priors in the scene during the learning process. The key contribution of this work is in two different places of the navigation pipeline: (1) an active learning setting for training the mapper that optimizes selection of maximally informative regions of the environment’s state space (metric locations on the floor-plan) for the agent to navigate towards and (2) incorporating the uncertainty in the mapper’s predictions while selecting the target semantic object on the map learned so far. The novel insight that connects both these contributions is borrowed from literature on intrinsic rewards via disagreement — the variance between the predictions of an ensemble of multiple mappers serves as a proxy for the model’s uncertainty about a specific location / the informativeness of that location for training the mapper.

**Summary Of The Review:**

As mentioned above, I feel that the paper has sufficient novel contributions to warrant an acceptance. I encourage the others to include a discussion on comparisons with SoTA from the leaderboard. Also, there are some minor issues that I hope get resolved as part of the discussions. But overall, keeping in mind the insights, novelty and usefulness of the ideas presented, my vote is to accept.

---

> ### Author Response · Authors · 2021-11-18
> **Reply to reviewer CF2f**
>
> Q: Some clarification on how the overall system works would be helpful in this context.
>
> A:  To clarify, the mapper is first trained with data collected by generating episodes using ground truth shortest paths to objects in different scenes. The mapper is then improved through fine-tuning with the active training procedure described in Section 3.2. The fully trained mapper is then used to select target semantic goal locations during testing. In summary, a) happens first during the training of the mapper which is used during testing which executes the policy in b).
>
> Q: The authors mention that the optimizations (eqn. 1-4), in practice, happen over the geo-centric map. Does it happen only for the set of map locations observed so far? That is, are there any constraints on the l_j’s in the geo-centric map?
>
> A: The locations are selected from any of the locations we have observed or any location we hallucinated with our mapping module so far. To be more specific, when we predict the semantic map outside of our field of view, we store or update the current predictions and observations of the agent in a global map. We update this map over the course of an episode with new observations and predictions. The long term goal is selected from this accumulated global map.
>
> Q: It is not immediately clear what is causing the variance in predictions for the ensemble of mappers. Is it because of different initializations? Or, is the training for each being warm-started on a different subset of the dataset?
>
> A: The variance between models comes from different random weight initializations. Using random weight initializations for diversity in ensembles of neural networks is commonly used in recent work (Pathak et al. 2019, Sekar et al. 2020), so we did not evaluate alternative ensemble training techniques. Furthermore, quoting Lakshminarayanan et al: “We found that random initialization of the NN parameters, along with random shuffling of the data points, was sufficient to obtain good performance in practice. We observed that bagging deteriorated performance in our experiments”.
>
> Sekar et al., Planning to explore via self-supervised world models, ICML 2020.
>
> Lakshminarayanan et al., “Simple and Scalable Predictive Uncertainty Estimation using Deep Ensembles”, NIPS 2017.
>
> Q: Inquiry over SPL performance and Offline-LowerBound method results
>
> A: We observe the difference in SPL (success weighted by path length) is smaller than the difference in success rate between the LowerBound and UpperBound. While the reviewer’s intuition is correct that for many navigation scenarios incentivizing exploration should lead to lower SPL given a fixed success rate, we recall that if an episode is not successful, SPL is zero. This observation comes directly from the formula for SPL for each episode which is given by (binary_success_indicator * shortest_path_length / taken_path_length). The difference in success rate impacts the difference in SPL, but since shorter paths are chosen for the LowerBound policy, the performance gap associated with exploration is smaller.
>
> Since “Active” outperformed “Offline” and “UpperBound” outperformed the other goal selection strategies, we intentionally only reported the “UpperBound” performance with the “Offline” model for brevity. We did perform those experiments and the results are:
>
> L2M-Offline-LowerBound (DTS:  3.733 +- 0.074, Success: 29.0 +- 0.9, SPL: 9.2 +- 0.4,Soft SPL: 15.0 +- 0.4)
>
> L2M-Offline-Mean (DTS: 3.690 +- 0.073, Success: 28.6 +- 0.9, SPL: 8.6 +- 0.3 Soft SPL: 14.9 +- 0.3)
>
> L2M-Offline-Mixed (DTS: 3.665 +- 0.072, Success: 28.6 +- 0.9, SPL: 9.0 +- 0.4, Soft SPL: 15.2 +- 0.4)
>
> Q: There are two rows for the same experiment in Tab.3 (L2M-Active-UpperBound). I suspect the reason is the special case for the SemExp baseline where the comparison happens only for a subset of 6 objects categories. Would be better to mention this explicitly in the text / table caption.
>
> A: Thank you for the suggestion! Your interpretation is correct. We will clarify in the final version.
>
> Q: Do they think that the same framework on variance of ensembles would work equally well in the semantic feature space (https://arxiv.org/abs/2010.01191)?
>
> A: Semantic hallucination and semantic segmentation have some technical differences which change the tradeoffs described in the paper cited by the reviewer. Hallucinating the semantic space is a hard problem, but hallucinating the RGB space is significantly harder. Therefore, we think evaluating variance between models in a hallucinated semantic space instead of a hallucinated RGB embedding will give a much better estimate of semantic prediction uncertainty. However, prior work has used the variance of ensembles in the latent space of models as an exploration bonus (https://arxiv.org/pdf/2005.05960.pdf). We thank the reviewer for their thoughtful comment, and we could investigate computing variance in the embedding space for our problem to ensure our intuition is correct.

---

### Official Review · Reviewer_Nmyk · 2021-11-04

**Correctness:** 2
**Technical Novelty And Significance:** 4
**Empirical Novelty And Significance:** 4
**Recommendation:** 5
**Confidence:** 4

**Main Review:**

### Strengths
- The method is well motivated and all components perform well
- The uncertainty component is used both for active learning and to select goals for ObjectNav
- Ablations are done well and comprehensive
- Code is provided

### Weaknesses

- Computational requirements. Uncertainty is estimated via an ensemble, thus increasing the computation by N(=4). Where other uncertainty estimate methods tried/considered?
- Lack of test-std results. The Habitat Challenge also contains a public leaderboard with a held-out test set (https://eval.ai/web/challenges/challenge-page/802/leaderboard/2195). The authors should submit to this.

### Suggestions for improvement

While I don't expect the authors to compare with these works as they are concurrent (one wasn't even public before the ICLR deadline), I encourage the authors also include Ye et al. (ICCV 2021) and Maksymets et al. (ICCV 2021) in their table for completeness.

Maksymets: https://openaccess.thecvf.com/content/ICCV2021/papers/Maksymets_THDA_Treasure_Hunt_Data_Augmentation_for_Semantic_Navigation_ICCV_2021_paper.pdf
Ye: https://arxiv.org/abs/2104.04112

**Summary Of The Paper:**

This paper presents a method to perform ObjectGoal navigation (i.e. goto the table) based on active semantic mapping. Specifically the proposed approach contains a semantic mapping module that hallucinates unseen areas. This mapping module is initially trained on a set of trajectories that are shortest paths between two points. Then additional trajectories are selected via active learning to maximize the learning signal.

ObjectGoal navigation is performed by training class specific map predictors and selecting goal locations via the upper confidence bound.

This method outperforms prior map-based works (SemExp) on the Habitat Challenge ObjectGoal navigation dataset.

**Summary Of The Review:**

This paper presents a strong contribution. My main concern is lack of test set results, however I consider this to be a significant issue as the test set is the standard for reporting on this task/dataset.

---

> ### Author Response · Authors · 2021-11-18
> **Reply to reviewer Nmyk**
>
> Q: Where other uncertainty estimate methods tried/considered?
>
> A: Yes, we considered using Bayesian neural networks to estimate uncertainty such as the approach described by Yarin Gal (https://arxiv.org/pdf/1506.02142.pdf). We decided to pursue the ensemble-based methods due to the arguments from Lakshiminarayanan (https://arxiv.org/pdf/1612.01474.pdf) which we found pertinent to our problem. We also  considered the tradeoffs between Bayesian methods and ensembles presented in this work (https://arxiv.org/pdf/2107.03342.pdf), particularly in Table 1. From the Lakshiminarayanan reference, the points most applicable to our problem include:
> 1. Ensembles are only more memory intensive. Otherwise, they can be comparably or even less computationally expensive than Bayesian neural networks. Specifically, the multiple passes required by Bayesian methods for every prediction can result in a higher computational cost than small ensembles.
> 2. Ensembles are highly parallelizable enabling which we leveraged during training, significantly reducing the cost of higher memory requirements.
> 3. In comparison to Bayesian methods, ensembles are simple to implement with little hyperparameter tuning which we found important when evaluating the use of ensembles versus Bayesian methods in a complex pipeline which already includes many other hyperparameters outside of the prediction model.
> 4. Ensembles are found to have equal or better performance in estimating uncertainty than Bayesian methods. (Sections 3.4, 3.5, 3.6)
>
> Gawlikowski et al., “A Survey of Uncertainty in Deep Neural Networks”, arXiv, 2021
> Lakshminarayanan et al., “Simple and Scalable Predictive Uncertainty Estimation using Deep Ensembles”, NIPS 2017.
>
> In addition to these points, we were happy to find when implementing our model that small ensemble sizes gave good results for uncertainty estimation. We also observed that averaging over an ensemble of map predictors increased the prediction performance which was not unexpected. These points supported our decision to use ensembles.
>
> Q: This paper presents a strong contribution. My main concern is lack of test set results, however I consider this to be a significant issue as the test set is the standard for reporting on this task/dataset.
>
> A: We thank the reviewer for their compliments on the contribution of our work. We understand their desire to see our results on the Habitat Challenge leaderboard. We tested a much earlier version of our model than the one presented in the paper on the 2021 Habitat Challenge (Clueless-Wanderers - PeterBot): https://eval.ai/web/challenges/challenge-page/802/leaderboard/2195. We plan to compete in the 2022 Habitat Challenge with our current model retrained on the Habitat configuration used in the challenge this year since the leaderboard is restarted every year with changes to the configuration. However, we do not believe that reporting results on local evaluation outside of this annual leaderboard should prevent publication. We list below several peer reviewed papers accepted for publication on this topic since the establishment of the leaderboard which do not evaluate on the leaderboard.
>
> https://arxiv.org/pdf/2006.10034.pdf (NeurIPS 2020)
>
> https://sscnav.cs.columbia.edu/ (ICRA 2021)
>
> https://arxiv.org/pdf/2106.04531.pdf (ICCV 2021)
>
> https://arxiv.org/pdf/2109.02066.pdf (ICCV 2021)

---

> > ### Comment · Reviewer_Nmyk · 2021-11-22
> > **Test-standard leaderboard**
> >
> > The test-standard split of the leaderboard for the habitat-challenge is open year round, only the test-challenge split/phase is run once per year.
> >
> > For the citations listed, only SSCNav uses the Habitat Challenge ObjectNav dataset. This is something that should have been caught in review of that paper but wasn't. The works by Ye et al and Maksymets et al I pointed to in my initial review do submit to the leaderboard.

---

> > > ### Author Response · Authors · 2021-11-22
> > > **Response to "Test-standard leaderboard"**
> > >
> > > Beyond SSCNav, we cite three more current papers in conferences (NeurIPS and ICCV) presenting ObjectNav results without having submitted to the challenge. These additional papers do not use the Habitat simulator and as a result do not compare to the methods deemed state of the art from the Habitat leaderboard. We do not believe that our work should be penalized in comparison to these works for using the Habitat simulator which we do in an effort to move toward standardized evaluation in the field. We appreciate all the effort invested in creating the challenge, and we plan to submit. We still do not believe it is or should be a gateway to publication considering the multiple published works on this topic which do not even use the same simulation environment.

---

> > > > ### Comment · Reviewer_Nmyk · 2021-12-02
> > > > **Test leaderboards**
> > > >
> > > > The standard practice in the ML/Vision/NLP communities is that when a dataset does have a test-set held behind an evaluation server, authors submit to this test server. The existence of datasets without a test-set behind an evaluation server (even when it's the same task) doesn't change this. I don't plan on blocking publication due to this issue as I think what's presented is interesting, but I encourage the authors to submit to the test leaderboard if for no other reasoning than increasing the visibility of the work.

---

> > > > > ### Author Response · Authors · 2021-12-06
> > > > > **Response to "Test leaderboards"**
> > > > >
> > > > > Thank you for agreeing to not block publication of our work. We appreciate the creation of the challenge and are working on our submission.

---

### Author Response · Authors · 2021-11-18
**Reply to all.**

We thank all the reviewers for their thoughtful and constructive comments. We particularly appreciate their questions and insights regarding the semantic map prediction model. We did our best to address the questions raised and we plan to include any new results in the final version of the paper.

---

### Decision · Program_Chairs · 2022-01-20

**Decision:**

Accept (Poster)

**Comment:**

This paper addresses the problem of goal navigation in unseen environments by learning to build a local, then a registered, global occupancy and semantic map of object categories from reprojected RGB+D observations, while extrapolating (hallucinating) unseen observations from contextual semantic priors (e.g., "tables are usually surrounded by chairs"). It then uses a measure of epistemic uncertainty on different estimations (realisations) of that map as a navigation goal selection policy to perform active exploration, and controls the agent using a local goal-driven policy; different information gain metrics are investigated. Essentially, the policy accumulates the predicted semantic maps and uses the uncertainty of the semantic mapping to select informative goals. The semantic map predictor is implemented as three U-nets for occupancy extrapolation from depth projection, semantic segmentation of RGB, and semantic map inference from the ground projection and extrapolated depth projection maps. The whole method is evaluated on the Matterport3D environment.

Reviewers praised the well-written and comprehensively-evaluated study, the active learning formulation and the idea of epistemic information gain as a measure of uncertainty for goal selection, and the code availability. Reviewers' major concerns included the computational cost of ensemble-based uncertainty estimation (Nmyk), and a missing submission to an active learderboard of the habitat-challenge (Nmyk, CF2f). Reviewers CF2f and qy8x had a longer list of issues that have been addressed in the rebuttal.

Reviewers engaged in a discussion with the authors, and the scores are 5, 6 (though not updated) and 8. I believe that the paper just meets the conference acceptance bar and would advocate for its inclusion in the conference.